# SCALING OMNI-MODAL PRETRAINING WITH MULTI-MODAL CONTEXT: ADVANCING UNIVERSAL REPRESENTATION LEARNING ACROSS MODALITIES

## ABSTRACT

In this work, we introduce Multimodal Context (MiCo), a scalable pretraining framework designed to advance omni-modal intelligence—an AI system capable of understanding and learning from multiple modalities to achieve universal representation learning. MiCo allows for efficient scaling of both the number of modalities and the volume of data, along with model parameters, during the pretraining phase. We evaluate the pretrained models across a diverse set of tasks, including: (i) single-modality perception benchmarks covering 10 distinct modalities, (ii) 25 cross-modal tasks spanning retrieval, question-answering, and captioning, and (iii) 18 large-scale multimodal language model benchmarks. MiCo consistently delivers state-of-the-art results, setting 37 new benchmarks across these tasks. The pretrained models, along with the collected datasets and codebase, will be made publicly available to support the development of omni-modal intelligence and broader research in multimodal learning.

## 1 INTRODUCTION

In the development of artificial intelligence, scalable pre-training has emerged as a promising pathway towards general intelligence (Radford et al., 2019; OpenAI, 2023; Brown et al., 2020; Radford et al., 2021; Bubeck et al., 2023). Additionally, pre-training has been established as an effective approach for learning more general and transferable representations across various modalities. For example, CLIP (Radford et al., 2021) constructs million-scale text-image pairs for cross-modal contrastive learning, making it one of the most impactful foundation models in the community (Rombach et al., 2022; Poole et al., 2022). Researchers have further extended the capabilities of CLIP (Radford

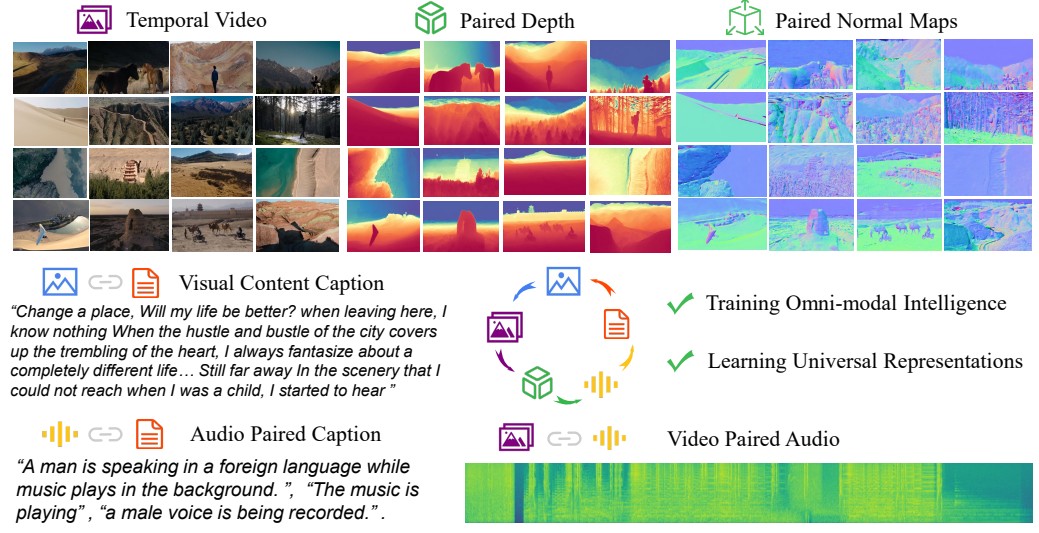

**Figure 1: Omni-modal Pretraining**. We propose collecting large-scale omni-modal paired data, including text, image, video, depth, and normal maps, to learn universal representations.

et al., 2021) to more data modalities, *e.g.* audio (Guzhov et al., 2022), point clouds (Xue et al.,

2023), and more comprehensive tasks, *e.g.* reasoning about images/ videos with large language models (LLMs) (Liu et al., 2024; Li et al., 2023d). The main contributions of CLIP (Radford et al., 2021) are two-fold: collecting web-scale text-image data and proposing a scalable vision-language pretraining paradigm. As more modalities *e.g.* audio, video, and 3D content, are getting widely used in this multimodal era (Han et al., 2023a; Li et al., 2023d; Ding et al., 2023; Girdhar et al., 2023; Rombach et al., 2022; Poole et al., 2022), such developments present additional challenges, including multimodal misalignment, misinterpretation, and bias amplification, in achieving coherent multimodal understanding with LLMs.

In this paper, we aim to enhance the comprehensive abilities of CLIP in visual understanding and further bolster its multimodal capacities across audio, video, 3D content, and more, as illustrated in Figure 1. This is significantly challenging. Therefore, we shift our focus from training a general multimodal model to understanding how the human brain performs coherent multimodal cognition. As outlined in Richard Mayer's Cognitive Theory of Multimedia Learning (Mayer, 2002), our brain processes multimedia signals through two distinct channels—auditory and visual—in sensory memory, as depicted in Figure 2. The sensory memory integrates these signals with prior knowledge through words, transforming new multimedia information into long-term memory. Notably, **1)** multimedia signals in the brain share channels, and **2)** words function as the reasoning interface in our brain.

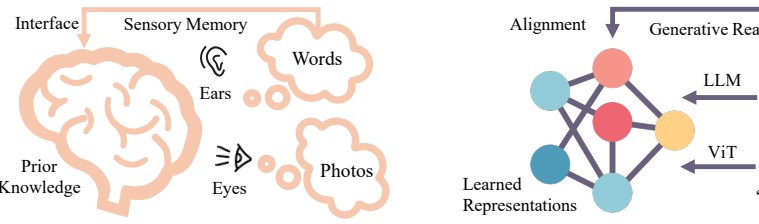

(a) Dual-Channel Multimodal Cognition Theory  (b) Brain-Inspired Omni-modal Learning Architecture

**Figure 2: Multimedia Cognition Process in Brain Inspires our Design**. We split diverse modalities into two types and employ individual neural networks to learn representations from each type respectively.

Inspired by these insights, we categorize diverse modalities into two types: "knowledge modality" and "interface modality". *Knowledge modalities*, primarily derived from raw sensors, contribute knowledge in diverse formats. For example, images and depth maps offer visual knowledge, while audio and video provide auditory and spatiotemporal knowledge. The language modality, developed by humans, is inherently more abstract and naturally functions as the *interface modality*, facilitating learning, reasoning, and the coordination of knowledge. To this end, we design an omni-modal learning architecture, illustrated in Figure 2 (b), with two distinct branches: one for knowledge modalities and one for the interface modality, *i.e.* natural language. The knowledge and interface modalities are aligned through a novel generative reasoning method, as detailed in § 3.3.

In addition to the architecture design, the next challenge is how to further enhance the benefits of integrating multiple data modalities. In Transformer (Vaswani et al., 2017), context relationship assigns a unique vector to each input position in a sequence and improves sequence modeling by capturing the sequential relationship among tokens. Moreover, since different modalities (*e.g.*, text, image, audio) offer complementary information, integrating these sources fosters a more comprehensive understanding of the data. Modeling token sequences from different modalities under the same context can help the model understand modality characteristics and joint semantics.

Therefore, we propose the Multimodal Context (MiCo) framework. We first map different modalities into a joint embedding space by sharing backbone networks. Then we build contextual relationships by joint context embeddings to enhance coherent multimodal understanding, as shown in Figure 1. Subsequently, we employ omnimodal contrastive learning, omnimodal feature matching, and omnimodal caption generation processes for pretraining (detailed in § 3.4). Moreover, MiCo can incorporate existing text-image, text-audio, and text-video datasets for joint multimodal context learning (§ 3.3), which leads to better omni-modal learning capacity, further modality extensibility, and easier scalability of multimodal data. Meanwhile, we explore the stability of MiCo in pretraining modalities, model parameters, and data scales (detailed in Figure 6).

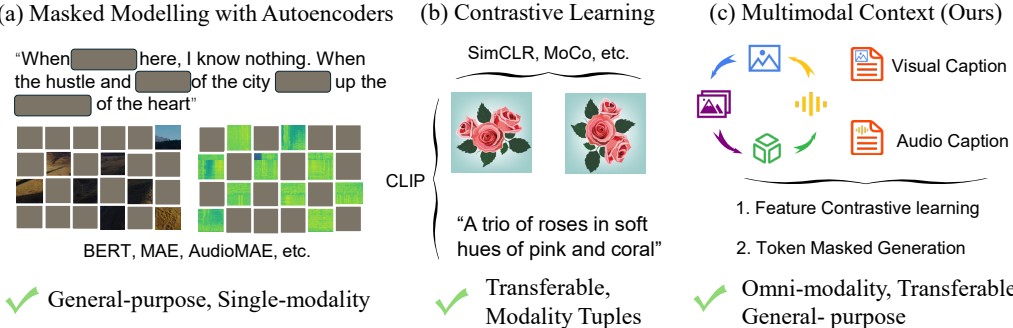

**Figure 3: Evolution of Pretraining Paradigms**. Masked modeling (He et al., 2022; Huang et al., 2022; Devlin et al., 2019) has shown great success in single-modality general-purpose understanding. Contrastive learning (He et al., 2020; Radford et al., 2021; Chen et al., 2020) distinguishes transferable features with modality tuples. We aim to achieve general-purpose omni-modal understanding and learn transferable, universal representations.

As shown in Figure 3, we compare MiCo with existing pretraining approaches. With omnimodal contrastive learning, omnimodal feature matching, and omnimodal caption generation processes, MiCo successfully integrates the advantages of both masked modeling and contrastive learning. In other words, MiCo represents the next-generation evolution of masked modeling (He et al., 2022; Huang et al., 2022; Devlin et al., 2019) and contrastive learning methods (He et al., 2020; Radford et al., 2021; Chen et al., 2020) for the multimodal era, offering significant benefits in omni-modal learning, strong transferability, and general-purpose representations. To thoroughly evaluate the effectiveness of MiCo, we conduct extensive experiments on universal single-modality perception benchmarks, cross-modal retrieval, captioning, and question-answer (QA) benchmarks, as well as zero-shot QA benchmarks for multimodal large language models. MiCo achieves impressive results across these benchmarks, establishing more than **37** new state-of-the-art (SOTA) performances and showing remarkable improvements of over **20%** on some benchmarks. These results compellingly illustrate that MiCo is a promising next-generation pretraining paradigm for the multimodal era.

## 2 RELATED WORK

**Vision-Language Pretraining.** MCAN (Yu et al., 2019b) first aligns vision and language features by stacking deep cross-attention blocks. Then more works (Wang et al., 2021b;c; 2022b;c) scale their models and improve the vision-language fusion process to build better alignment. VL-BERT (Su et al., 2019) introduced the Masked Language Model (MLM) paradigm, focusing on generic tasks across both vision and language modalities. Then Oscar (Li et al., 2020) proposed to enrich the representation of object semantics by integrating visual and textual content. Subsequent frameworks have further refined and extended these capabilities. Notably, VinVL (Zhang et al., 2021), SimVLM (Wang et al., 2021c), VLMO (Wang et al., 2021b), ALBEF (Li et al., 2021a), and Florence (Yuan et al., 2021) have explored and demonstrated the advantages of joint representations that ensure semantic consistency across the visual and natural language. Additionally, the versatility of multimodal models extends into specialized applications such as few-shot learning (Alayrac et al., 2022), and sequence-to-sequence (Wang et al., 2022b; Yu et al., 2022). BEiT-v3 (Wang et al., 2022c) employs a cross-modal mask-and-reconstruction process with partially shared parameters.

**More-Modality Pretraining**. MMV (Alayrac et al., 2020) pioneered multimodal pretraining using text, video, and audio pairs. They proposed multimodal contrastive learning for alignment. Then VATT (Akbari et al., 2021) further developed pretraining multiple modalities with transformers. After CLIP (Radford et al., 2021), more works (Zhang et al., 2023c; Girdhar et al., 2023; Guzhov et al., 2022; Xue et al., 2023; Xu et al., 2021; Wang et al., 2024) propose to adapt pretrained CLIP models to more modalities including point cloud, depth, audio, video, *etc*. Another direction is to exploit multimodal complementary benefits and construct more modality pairs such as VAST (Chen et al., 2023b) and VALOR (Chen et al., 2023a), which improve the abilities for multimodal understanding.

Despite significant advancements in multimodal learning, several key challenges impede the development of comprehensive omni-modal intelligence: **1) Focus on Vision-Language**: Current methods (Wang et al., 2022c; 2021b; Li et al., 2020; Wang et al., 2021c) predominantly cater to vision-language tasks. The inflexibility of these works limits the extension with more modalities such

as video, audio, *etc*. **2**) **Architectural Constraints**: The development of architectures capable of handling a broader array of modalities is still in its nascent stages. *Crafting scalable and efficient multimodal learning architectures presents a significant challenge.* **3**) **Data Availability**: There is a notable scarcity of publicly accessible datasets including multimodal paired data (video, depth, audio, and captions). **4**) **Multimodal Benefits**: Although leveraging the synergistic benefits of multiple modalities is crucial (Fei et al., 2022), *understanding and optimizing the interaction between highly disparate modalities remains a complex and largely unexplored area.*

## 3 MULTIMODAL CONTEXT

### 3.1 LARGE-SCALE DATA COLLECTION

We use the HD-VILA (Xue et al., 2022) dataset, which contains 371.5K hours of 720p ($1280 \times 720$) videos. We remove video clips that are shorter than 5s or longer than 30s. Then, we collect a dataset containing 1.7M paired video clips ($\sim$510M frames), audio, and subtitles $\{(\boldsymbol{x}_V, \boldsymbol{x}_T^V, \boldsymbol{x}_A)\}$. Then we enrich the dataset by adding captions to video frames (images), and audio with pre-trained captioners (Chen et al., 2023b), getting $(\boldsymbol{x}_I, \boldsymbol{x}_T^I)$ and $(\boldsymbol{x}_A, \boldsymbol{x}_T^A)$. Finally, we use pre-trained monocular depth estimation models (Fu et al., 2024; Eftekhar et al., 2021)[1] to generate depth and normal maps, getting $(\boldsymbol{x}_I, \boldsymbol{x}_D, \boldsymbol{x}_N)$. Thus, we collect million-scale multimodal paired data $\{(\boldsymbol{x}_I, \boldsymbol{x}_D, \boldsymbol{x}_N, \boldsymbol{x}_T^I), (\boldsymbol{x}_A, \boldsymbol{x}_T^A), (\boldsymbol{x}_V, \boldsymbol{x}_T^V)\}$, where $\boldsymbol{x}_T, \boldsymbol{x}_I, \boldsymbol{x}_A, \boldsymbol{x}_V$, and $\boldsymbol{x}_D$ denote the modality-specific samples of text captions, image, audio, video clips, depth, and normal maps. We split our dataset into several subsets including 1M, 10M, 110M, and 334M multimodal data pairs, and we provide detailed illustrations in Appendix C.

### 3.2 ARCHITECTURE DESIGN FOR OMNI-MODAL LEARNING

We first investigate several variants of encoder architectures with four data modalities. With our collected data, we pretrain architectures for 300K steps by the same contrastive (Radford et al., 2021) and masked-generation loss functions (Wang et al., 2022c) (details in Appendix B). We take the captioning and retrieval tasks on image, audio, and video modalities as the main evaluation benchmark for designing architectures.

**Architectural Designs**. We construct the vanilla architecture from CLIP (Radford et al., 2021). A text encoder of Transformer (Vaswani et al., 2017) takes text inputs and outputs text embeddings $\boldsymbol{z}_T$, and an image encoder of Vision Transformer (Dosovitskiy et al., 2021) takes image input $x_I \in \mathbb{R}^{3 \times H \times W}$ and outputs image embeddings $\boldsymbol{z}_I$, respectively.

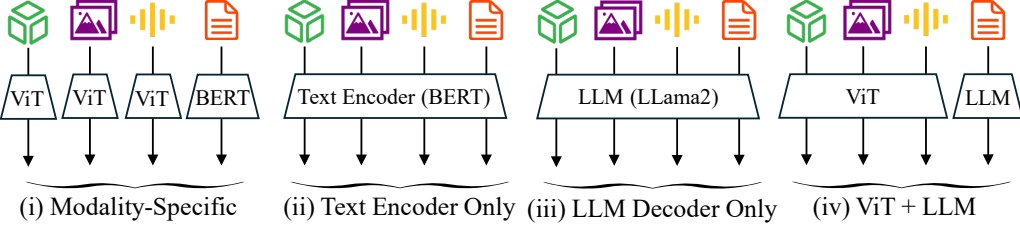

(i) Modality-Specific    (ii) Text Encoder Only  (iii) LLM Decoder Only    (iv) ViT + LLM

**Figure 4:** Options of Architecture Design for Omni-Modal Pretraining.

As shown in Figure 4, we propose 4 architectures for omni-modal learning: **i**) Modality-specific encoders for each modality, employing individual transformers to extract multimodal embeddings, then fuse them as BEiT-3 (Wang et al., 2022c). **ii**) BERT (text encoder) as a unified multimodal encoder to extract multimodal embeddings and generates texts. **iii**) LLM (text decoder) as a unified multimodal encoder and text generator. **iv**) A ViT as a unified multimodal encoder besides text, and an LLM deals with text embeddings and generation.

**Empirical Discovery.** Referring to Table 1, we conclude that: **1**) *Pure language models are difficult to retrieval tasks*. Both (ii) and (iii) deliver a significant performance drop in retrieval tasks. **2**) *No more than 2 Encoders*. Comparing (i) with (ii) & (iii), we observe that additional encoders are

---

[1]Geowizard (Fu et al., 2024) delivers significantly better annotations, while DPT (Eftekhar et al., 2021) predicts much faster (about $34.7\times$ faster). We use the Geowizard to annotate the high-quality data about 2M.

**Table 1: Architecture Design of Omni-Modal Learning Paradigm**. We the ViT-g and Llama-2-7B (Touvron et al., 2023) in this table. We pretrain models for 300k steps, then evaluate performances on the MSRVTT, VATEX, AudioCaps, ClothoV2, COCO, and Flicker datasets for caption (CIDEr) and retrieval tasks (R@1).

| Architecture | Video | | Audio | | Image | |
|---|---|---|---|---|---|---|
| | MSRVTT | VATEX | AudioCaps | ClothoV2 | COCO | Flickr |
| | CIDEr (%) | R@1 (%) | R@1 (%) | CIDEr (%) | R@1 (%) | R@1 (%) |
| (i) Modality-Specific | 74.3 | 73.5 | 42.3 | 22.3 | 65.2 | 88.4 |
| (ii) Text Encoder (BERT) | 77.0 | 53.2 | 23.1 | 43.9 | 46.7 | 51.6 |
| (iii) LLM (LLama-2-7B) | 75.2 | 60.3 | 14.7 | 43.6 | 60.8 | 81.3 |
| (iv) ViT + LLM | **77.9** | **79.5** | **49.7** | **47.2** | **67.5** | **90.5** |

beneficial for retrieval tasks; however, a comparison between (i) and (iv) suggests that discrepancies among multiple encoders can also hinder multimodal alignment. **3)** *Language is an individual branch for alignment.* Comparing (ii) & (iii), with (iv), improvements are significant in both retrieval and captioning.

### 3.3 MULTIMODAL CONTEXT CONSTRUCTION

**Preliminary.** The context is proposed to assign a unique vector to each token in a sequence (Vaswani et al., 2017), which reinforces potential relevance between positions. Different modalities (*e.g.*, text, image, audio) provide complementary information. Learning multimodal context leads to a more holistic and nuanced understanding of data. It can also leverage the strengths of each modality and guide the model to understand the interactions between different types of information. Therefore, we seek to construct the context relationship across diverse modalities and extend the learning capacity to omni-modalies. We provide the overview of MiCo pretraining paradigm in Figure 5.

**Single Dataset with Multimodal Paired Data.** As mentioned in § 3.1, we build a dataset with multimodal paired data $\{(\boldsymbol{x}_I, \boldsymbol{x}_D, \boldsymbol{x}_N, \boldsymbol{x}_T^I), (\boldsymbol{x}_A, \boldsymbol{x}_T^A), (\boldsymbol{x}_V, \boldsymbol{x}_T^V)\}$, then we employ the omni-modal encoder $f(\cdot; \theta)$ to extract features $\boldsymbol{z}_I, \boldsymbol{z}_A, \boldsymbol{z}_V, \boldsymbol{z}_D$, and $\boldsymbol{z}_N$, then use text encoder to extract text features $\boldsymbol{z}_T$. Therefore, we construct the context by a top-down design: **1)** For the whole multimodal embeddings, they share the same position embeddings $\boldsymbol{E}_{\text{Pos}}$ to build a modality-fused context relationship across diverse modalities. **2)** Then, for each specific context, they're labeled by modality embeddings including $\boldsymbol{E}_M^I, \boldsymbol{E}_M^A, \boldsymbol{E}_M^V, \boldsymbol{E}_M^D, \boldsymbol{E}_M^N$, *etc* to indicate modality types. **3)** Within the same modality context, we employ the context embeddings $\boldsymbol{E}_C^I$ to construct uni-modal context relationships. Thus, the construction of the multimodal context can be formulated as:

$$\boldsymbol{z}_I = [\boldsymbol{z}_I^1, \boldsymbol{z}_I^2, \cdots, \boldsymbol{z}_I^{L_I}] + \boldsymbol{E}_C^I, \quad \text{for each modality,}$$
$$\boldsymbol{z} = [\boldsymbol{z}_I + \boldsymbol{E}_M^I, \boldsymbol{z}_A + \boldsymbol{E}_M^A, \boldsymbol{z}_V + \boldsymbol{E}_M^V, \boldsymbol{z}_D + \boldsymbol{E}_M^D, \boldsymbol{z}_N + \boldsymbol{E}_M^N] + \boldsymbol{E}_{\text{Pos}}, \tag{1}$$

where $\boldsymbol{E}_C^I$ is up to the sample length of a specific modality. Meanwhile, the text features of specific captions can be easily concatenated, where their position embeddings $\boldsymbol{E}_{\text{Pos}}'$ are also shared:

$$\boldsymbol{z}_T = [\boldsymbol{z}_T^I, \boldsymbol{z}_T^A, \boldsymbol{z}_T^V] + \boldsymbol{E}_{\text{Pos}}'. \tag{2}$$

**Multiple Datasets Combination of Cross-Modal Datasets.** Besides multimodal paired data, our proposed paradigm can also leverage existing web-scale text-image, text-audio, and text-video datasets to jointly pretraining models towards omni-modal universal representations. Given datasets $\mathcal{D}_I = \{(\boldsymbol{x}_I^j, \boldsymbol{x}_T^j)\}_{j=1}^{N_I}, \mathcal{D}_A = \{(\boldsymbol{x}_A^j, \boldsymbol{x}_T^j)\}_{j=1}^{N_A}$, and $\mathcal{D}_V = \{(\boldsymbol{x}_V^j, \boldsymbol{x}_T^j)\}_{j=1}^{N_V}$, each pair of data possess local and simple context, for example, a pair of text-image data $(\boldsymbol{x}_I, \boldsymbol{x}_T)$ corresponds to a simple context $(\boldsymbol{z}_I + \boldsymbol{E}_{\text{Pos}}, \boldsymbol{E}_{\text{Pos}}')$, which may limit the learned representations of models. We propose to build the multimodal context by cross-dataset joint sampling with sampling context embedding $\boldsymbol{E}_{\text{Sam}}$:

$$(\boldsymbol{x}_I, \boldsymbol{x}_T^I) = \text{Sample}(\mathcal{D}_I), \ (\boldsymbol{x}_A, \boldsymbol{x}_T^A) = \text{Sample}(\mathcal{D}_A), \ (\boldsymbol{x}_V, \boldsymbol{x}_T^V) = \text{Sample}(\mathcal{D}_V),$$
$$\boldsymbol{z}_I = f(\boldsymbol{x}_I; \theta) + \boldsymbol{E}_{\text{Sam}}^{T-I}, \ \ \boldsymbol{z}_T^I = f'(\boldsymbol{x}_T; \theta') + \boldsymbol{E}_{\text{Sam}}^{T-I}, \quad \text{for each modality,} \tag{3}$$
$$\boldsymbol{z} = [\boldsymbol{z}_I + \boldsymbol{E}_M^I, \boldsymbol{z}_A + \boldsymbol{E}_M^A, \boldsymbol{z}_V + \boldsymbol{E}_M^V] + \boldsymbol{E}_{\text{Pos}}, \ \ \boldsymbol{z}_T = [\boldsymbol{z}_T^I, \boldsymbol{z}_T^A, \boldsymbol{z}_T^V] + \boldsymbol{E}_{\text{Pos}}'.$$

In this way, we successfully combine existing multiple cross-modal datasets towards learning omni-modal universal representations by building more universal and complicated multimodal contexts (Equation 3) for pretraining models, therefore, *MiCo can outperform existing pretraining methods by better generalization learning ability, modality extensibility, and easier for scaling data.*

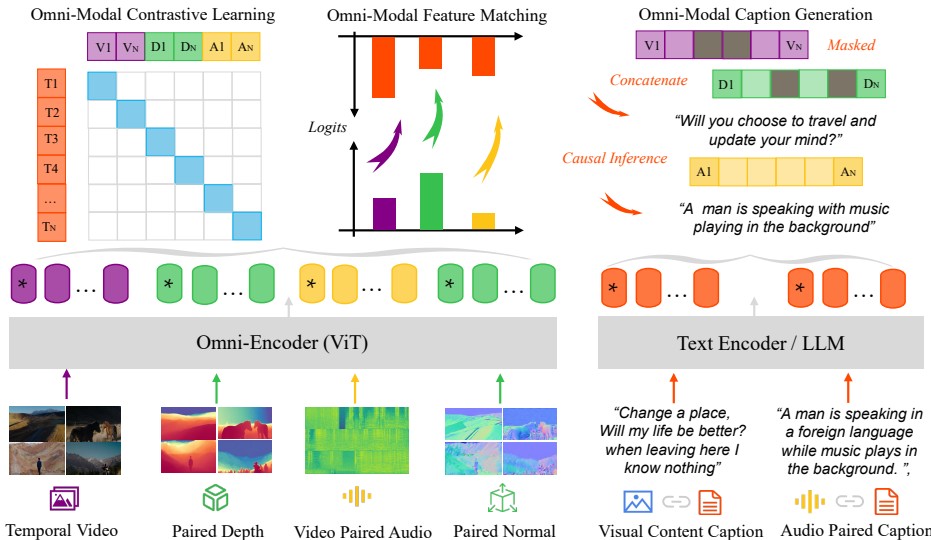

**Figure 5: Overview of Multimodal Context Pretraining Paradigm**. We use a shared ViT for multimodal feature extraction, and another branch is to employ a text encoder. We concatenate these multimodal sequences as multimodal contexts and perform contrastive learning and masked modeling.

## 3.4 PRETRAINING OBJECTIVES

***Omni-modal Contrastive Learning.*** The omni-modality representations are denoted as $z$. Subsequently, $z$ and $z_T$ are projected into the same space using MLPs. The omni-modal contrastive learning is formulated by the dot product of $z$ and $z_T$. We use $v^z$ and $v^T$ to denote projected vectors:

$$\mathcal{L}_{\text{Con}} = -\frac{1}{2} \sum_{i=1}^{N_B} \log \frac{\exp(\tau \cdot < v_i^z, v_i^T >)}{\sum_{j=1}^{N_B} \exp(\tau \cdot < v_i^z, v_j^T >))} - \frac{1}{2} \sum_{i=1}^{N_B} \log \frac{\exp(\tau \cdot < v_i^z, v_i^T >))}{\sum_{j=1}^{N_B} \exp(\tau \cdot < v_j^z, v_i^T >))}, \quad (4)$$

where $< \cdot, \cdot >$, $N_B$ and $\tau$ denote the dot product, batch size, and a learnable parameter.

***Omni-modal Feature Matching Process*** is designed to improve the semantic alignment between multimodal (knowledge modalities) and textual features. We employ an MLP layer to perform binary predictions $p_v$ of $(z, z_T)$. Following a hard negative mining strategy(Li et al., 2021a), we assigns $y = 1$ if features are matched, and $y = 0$ otherwise.

$$\mathcal{L}_{\text{Match}} = \mathbb{E}_{(v_i^z, v_i^T) \sim (\mathcal{Z}, \mathcal{T})} \left[ y \log p_v + (1 - y) \log (1 - p_v) \right] \quad (5)$$

***Omni-modal Caption Generation Process.*** We employ conditional causal masked (60%) language modeling for generative omni-modal reasoning. In specific, a single-directional causal attention mask is used to avoid information leakage, and the masked tokens are reconstructed using a prediction layer of BERT (Devlin et al., 2019). We use $c_m$ and $c_{<m}$ to denote masked tokens and former tokens, respectively.

$$\mathcal{L}_{\text{Gen}} = -\mathbb{E}_{(v_i^T, v_i^T) \sim (\mathcal{V}, \mathcal{T})} \log P \left( c_m \mid c_{<m}, v^z \right) \quad (6)$$

## 4 EXPERIMENT

### 4.1 EXPERIMENTAL SETUP

**1**) Single-modality Understanding § 4.2 (following previous practices (Radford et al., 2021; Zhang et al., 2023c; Girdhar et al., 2023) in fine-tuning & zero-shot setting in classification and forecasting tasks), **2**) Cross-modality Understanding § 4.3 (following BEiT-3 (Wang et al., 2022c), VAST (Chen et al., 2023b) in fine-tuning and dataset splits for Caption, QA, and retrieval tasks), and **3**) Multimodal Understanding with Large Language Models § 4.4 (following LLava (Liu et al., 2023a), VideoChat (Li

et al., 2023f), OneLLM (Han et al., 2023a) in multimodal zero-shot QA). Detailed experimental settings including datasets introduction, splits, and evaluation metrics can be found in our Appendix C.

**Implementation Details.** We implement our pretraining paradigm with ViT backbone scaling from ViT-B (8 NVIDIA Tesla A100 GPUs) to ViT-g (128 NVIDIA Tesla A100 GPUs). It takes about 2~6 days for pretraining. We pretrain models 200k steps for ViT-B, 300k steps for ViT-L and ViT-g. The initial learning rate is set to 1e-4, and a linear decay schedule is used. The batch size on each GPU is set to 1,024. More implementation details can be found in the Appendix B.

**Table 2: State-of-the-art Abilities of MiCo for Omni-modal Perception**. We report the Accuracy (%) of MMLU (Hendrycks et al., 2020), IN-1K (Deng et al., 2009), K700 (Kay et al., 2017), NYU-D (Nathan Silberman & Fergus, 2012), Ego4D (Grauman et al., 2022), Indian Pines, and Fraud datasets, R@1 (%) for MSR-VTT (Xu et al., 2016b) and SYSU (Wu et al., 2017), mAP for AS-2M (Gemmeke et al., 2017), F1-score for Fraud, and Mean Absulte Error↓ for PCQM4M and Global Weather Forecasting (Wu et al., 2023) benchmarks. We detail tasks and previous SOTA methds in the Appendix.

| Methods (Backbone) | Text MMLU | Image IN-1K | Video K700/MSR-VTT | Depth NYU-D | Audio AS-2M | Thermal SYSU | IMU Ego4D | Graph PCQM4M | Time-Series Global Weather | Hyperspectral IP | Tabular Fraud |
|---|---|---|---|---|---|---|---|---|---|---|---|
| ImageBind (ViT-H) | 43.6 | 80.2 | 42.9/36.8 | 54.0 | 43.4 | 72.6 | 25.0 | 0.815↓ | 8.439↓ | 83.6 | 0.847 |
| Meta-Trans (ViT-L) | 37.3 | 88.1 | 33.2/31.5 | 41.5 | 38.9 | 71.3 | 73.9 | 0.886↓ | 7.892↓ | 78.1 | 0.809 |
| Absolute SOTA | 90.0 | 91.0 | 92.1/62.8 | 76.7 | 48.6 | 77.9 | 52.5 | 0.123 | 7.602↓ | 98.0 | 0.860 |
| MiCo (ViT-g) [Ours] | 68.9 | 89.8 | 91.6/**64.3** | **84.6** | **50.5** | **80.3** | **77.2** | 0.742↓ | 7.834↓ | **98.5** | **0.913** |

## 4.2 EVALUATION ON SINGLE-MODALITY UNDERSTANDING

**Exceptional Omni-modal Perception Abilities.** As shown in Table 2, MiCo achieves state-of-the-art performances on a range of benchmarks across 10 modalities. For text understanding (MMLU), MiCo attains the accuracy of 68.9%, outperforming both ImageBind (Girdhar et al., 2023) (43.6%) and Meta-Transformer (Zhang et al., 2023c) (37.3%). In image recognition (IN-1K), MiCo delivers Top-1 Acc. of 89.8%. On K700 and MSR-VTT, MiCo achieves 91.6% for Acc. and R@1 of 64.3%, outperforming existing retrieval methods. Regrading 3D singe-view tasks (NYU-D), MiCo outperforms the absolute SOTA (Girdhar et al., 2022) by +7.9%. On AS-2M, MiCo achieves the mAP of 50.5%, which is better than BEATS-3 (Chen et al., 2022a) by +1.9%. MiCo also excels in thermal sensing (SYSU) and IMU tasks (Ego4D), MiCo achieves an accuracy of 80.3% and 77.2%, respectively. *These results highlight MiCo's comprehensive and outstanding performances, establishing it as a powerful model for omni-modal perception.*

**Table 3: Powerful Cross-Modal Abilities**. We evaluate MiCo on the mainstream cross-modal tasks including 11 retrieval tasks (COCO (Lin et al., 2014), Flickr (Plummer et al., 2015), ClothoV1 (Drossos et al., 2020), ClothoV2 (Drossos et al., 2020), AudioCaps (Kim et al., 2019), MSRVTT (Xu et al., 2016a), YouCook2 (Zhou et al., 2018), VALOR-32K (Chen et al., 2023a), VATEX (Wang et al., 2019), DEDeMo (Anne Hendricks et al., 2017), and ANET (Yu et al., 2019a)), 7 caption tasks (COCO, ClothoV1, ClothoV2, AudioCaps, MSRVTT, YouCook2, VALOR-32K), and 6 QA tasks (TGIF (Li et al., 2016), MSVD (Xu et al., 2017), VQAv2 (Goyal et al., 2017a), MSRVTT, MUSIC (Li et al., 2022), and ANET) with the metrics of R@1, CIDEr, and Acc. Impressively, **MiCo archives 20 new SoTA performances**.

**Image**

| | Text-to-Image Retrieval | | | Image Caption | Visual QA | | |
|---|---|---|---|---|---|---|---|
| | COCO | Flickr | Flickr(ZS) | COCO | TGIF | MSVD | VQAv2 |
| SOTA | 68.3 (Li et al., 2023c) | 90.3 (Wang et al., 2022c) | 89.7 (Li et al., 2023c) | 154.9* (Wang et al., 2022a) | 78.7 (Chen et al., 2023a) | 60.2 (Kuo et al., 2023) | 84.3 (Chen et al., 2022b) |
| MiCo | 68.1 | **91.1** ↑0.8 | **90.1** ↑0.4 | 152.4 | **78.9** ↑0.2 | **60.4** ↑0.2 | 80.5 |

**Audio**

| | Text-to-Audio Retrieval | | | Audio Caption | | |
|---|---|---|---|---|---|---|
| | ClothoV1 | ClothoV2 | AudioCaps | ClothoV1 | ClothoV2 | AudioCaps |
| SOTA | 17.5 (Chen et al., 2023a) | 21.5 (Mei et al., 2023) | 42.2 (Mei et al., 2023) | 42.3 (Chen et al., 2023a) | 48.8 (Mei et al., 2023) | 78.7 (Mei et al., 2023) |
| MiCo | **21.2** ↑3.7 | **23.3** ↑1.8 | 41.0 | 41.0 | **49.6** ↑7.3 | **50.8** ↑2.0 | 66.2 |

**Video-Audio**

| | Text-to-Video-Audio Retrieval | | | | | |
|---|---|---|---|---|---|---|
| | MSRVTT | YouCook2 | VALOR-32K | VATEX | DiDeMo | ANET |
| SOTA | 54.4 (Chen et al., 2023a) | 31.3 (Li et al., 2021b) | 73.2 (Chen et al., 2023a) | 76.9 (Chen et al., 2023a) | 57.6 (Chen et al., 2023a) | 63.4 (Chen et al., 2023a) |
| MiCo | **64.3** ↑9.9 | **51.3** ↑20.0 | **78.7** ↑5.5 | **81.3** ↑4.4 | **63.6** ↑6.0 | **68.5** ↑5.1 |

**Video-Audio**

| | Video-Audio Caption | | | Video-Audio QA | | |
|---|---|---|---|---|---|---|
| | MSRVTT | YouCook2 | VALOR-32K | MSRVTT | MUSIC | ANET |
| SOTA | 74.0 (Chen et al., 2023a) | 190.0 (Ko et al., 2023) | 61.5 (Chen et al., 2023a) | 49.2 (Chen et al., 2023a) | 78.9 (Chen et al., 2023a) | 48.6 (Chen et al., 2023a) |
| MiCo | **79.3** ↑5.3 | **197.8** ↑7.8 | **62.8** ↑1.3 | **50.4** ↑1.2 | **79.7** ↑0.8 | **51.0** ↑2.4 |

## 4.3 EVALUATION ON CROSS-MODAL UNDERSTANDING

Table 3 illustrates the powerful performances of MiCo on 25 cross-modal benchmarks, **achieving more than 20 new SOTA performances**. For text-to-image retrieval, MiCo achieves outstanding results with R@1 of 68.1% on COCO, and 91.1% on Flickr, outperforming previous SOTA methods. For VQA, MiCo demonstrates robust performance with accuracy scores of 78.9% on TGIF, 60.4% on MSVD, and 80.5% on VQA v2, highlighting its strong visual comprehension and reasoning abilities. In text-to-audio retrieval, MiCo achieves outstanding performances of 21.2% on ClothoV1, 23.3%

**Table 4: Evaluation on LLM Benchmarks.** The MLLM evaluation involves 6 VQA tasks (GQA (Hudson & Manning, 2019), VQAv2 (Goyal et al., 2017b), OKVQA (Marino et al., 2019), TextVQA (TVQA) (Singh et al., 2019), ScienceQA (SQA) (Lu et al., 2022) and Vizwiz (Gurari et al., 2018)), 2 image captioning tasks (Nocaps (Agrawal et al., 2019) and Flickr30K (Plummer et al., 2015)), and 4 multimodal benchmarks (MME (Fu et al., 2023), MM Bench (MMB) (Liu et al., 2023c), MMVet (Yu et al., 2023) and SEED (Li et al., 2023a)). The evaluation metrics for VQA and captioning tasks are accuracy and CIDEr, respectively.

| Method | LLM | Visual Question Answering | | | | | | Image Caption | | MM Benchmark | | | |
|---|---|---|---|---|---|---|---|---|---|---|---|---|---|
| | | GQA | VQAv2 | OKVQA | TVQA | SQA | Vizwiz | NoCaps | Flickr | MME | MMB | MMVet | SEED |
| *Vision Specialist LLM* | | | | | | | | | | | | | |
| Flamingo-9B (Alayrac et al., 2022) | Chinchilla-7B | - | 51.8 | 44.7 | 30.1 | - | 28.8 | - | 61.5 | - | - | - | - |
| Flamingo-80B (Alayrac et al., 2022) | Chinchilla-70B | - | 56.3 | 50.6 | 31.8 | - | 31.6 | - | 67.2 | - | - | - | - |
| BLIP-2 (Li et al., 2023c) | Vicuna-7B | - | - | - | 40.1 | 53.8 | - | 107.5 | 74.9 | - | - | - | - |
| BLIP-2 (Li et al., 2023c) | Vicuna-13B | 41.0 | 41.0 | - | 42.5 | 61 | 19.6 | 103.9 | 71.6 | 1293.8 | - | 22.4 | - |
| InstructBLIP (Dai et al., 2023) | Vicuna-7B | 49.2 | - | - | 50.1 | 60.5 | 34.5 | 123.1 | 82.4 | - | 36 | 26.2 | - |
| InstructBLIP (Dai et al., 2023) | Vicuna-13B | 49.5 | - | - | 50.7 | 63.1 | 34.3 | 121.9 | 82.8 | 1212.8 | - | 25.6 | - |
| IDEFICS-9B (Laurençon et al., 2023) | LLaMA-7B | 38.4 | 50.9 | 38.4 | 25.9 | - | 35.5 | - | 27.3 | - | 48.2 | - | - |
| IDEFICS-80B (Laurençon et al., 2023) | LLaMA-65B | 45.2 | 60.0 | 45.2 | 30.9 | - | 36.0 | - | 53.7 | - | 54.5 | - | - |
| LLaMA-Ad.v2 (Gao et al., 2023) | LLaMA-7B | 43.9 | - | 55.9 | 43.8 | 54.2 | - | 42.7 | 30.5 | 972.7 | 38.9 | 31.4 | 32.7 |
| Qwen-VL (Bai et al., 2023) | Qwen-7B | 57.5 | 78.2 | 56.6 | 61.5 | 68.2 | 38.9 | 120.2 | 81.0 | 1487.5 | 60.6 | - | 58.2 |
| LLaVA-v1.5 (Liu et al., 2023a) | Vicuna-7B | 62.0 | 78.5 | - | 58.2 | 66.8 | 50.0 | - | - | 1510.7 | 64.3 | 30.5 | 58.6 |
| *Multimodal Generalist LLM* | | | | | | | | | | | | | |
| ImageBind-LLM (Han et al., 2023b) | LLaMA-7B | 41.1 | - | - | 24.0 | 51.4 | - | 29.6 | 23.5 | 775.7 | - | - | - |
| ChatBridge-13B (Zhao et al., 2023) | Vicuna-13B | 41.8 | - | 45.2 | - | - | - | 115.7 | 82.5 | - | - | - | - |
| AnyMAL-13B (Moon et al., 2023) | LLaMA2-13B | - | 59.6 | 33.1 | 24.7 | 52.7 | 24.4 | - | - | - | - | - | - |
| AnyMAL-70B (Moon et al., 2023) | LLaMA2-70B | - | 64.2 | 42.6 | 32.9 | 70.8 | 33.8 | - | - | - | - | - | - |
| OneLLM-7B [CVPR'24] | LLaMA2-7B | 59.5 | 71.6 | 58.9 | 34.0 | 63.4 | 45.9 | 115.9 | 78.6 | 1392.0 | 60.0 | 29.1 | 61.2 |
| **MiCo-Chat-7B** | Qwen2-7B | **66.5** | **79.5** | **59.6** | **63.4** | **77.5** | 49.1 | **128.5** | 79.8 | 1574.6 | 70.4 | 49.3 | 69.2 |

**Table 5: Zero-Shot Audio & Video generative benchmark with LLMs.** We evaluate models by audio captioning on Clotho Caption (Drossos et al., 2020), audio QA on Clotho AQA (Lipping et al., 2022) and video-based generative performance benchmark (Maaz et al., 2023) using the same Vicuna-7B.

| Method | 0-shot | Clotho Caption | | Clotho AQA |
|---|---|---|---|---|
| | | CIDEr | SPIDEr | Acc. |
| FeatureCut (Ye et al., 2022) | ✗ | 43.6 | 27.9 | - |
| Wavcaps (Mei et al., 2023) | ✗ | 48.8 | 31.0 | - |
| MWAFM (Li et al., 2023b) | ✗ | - | - | 22.2 |
| Pengi (Deshmukh et al., 2023) | ✗ | - | 27.1 | 64.5 |
| ChatBridge-13B (Zhao et al., 2023) | ✓ | 26.2 | - | - |
| OneLLM-7B | ✓ | 29.1 | 19.5 | 57.9 |
| **MiCo-Chat-7B** [Ours] | ✓ | **33.3** | **21.9** | 63.9 |

| Method | Cor. | Det. | Con. | Tem. | Cons. |
|---|---|---|---|---|---|
| VideoLLaMA (Zhang et al., 2023b) | 1.96 | 2.18 | 2.16 | 1.82 | 1.79 |
| VideoChat (Li et al., 2023e) | 2.23 | 2.50 | 2.53 | 1.94 | 2.24 |
| Video-ChatGPT (Maaz et al., 2023) | 2.40 | 2.52 | 2.62 | 1.98 | 2.37 |
| BT-Adapter (Liu et al., 2023b) | 2.68 | 2.69 | 3.27 | 2.34 | 2.46 |
| LLaMa-VID (Li et al., 2023g) | 2.96 | 3.00 | 3.53 | 2.46 | 2.51 |
| **MiCo-Chat-7B** [Ours] | **3.00** | **3.01** | **3.61** | **2.49** | **2.71** |

**Table 6: Zero-shot Video QA with LLMs.** In comparison with leading methods, we report results with 1 token for each frame, where Res. indicates image resolution.

| Method | LLM | Res. | MSVD-QA | | MSRVTT-QA | | ActivityNet-QA | |
|---|---|---|---|---|---|---|---|---|
| | | | Acc | Score | Acc | Score | Acc | Score |
| FrozenBiLM (Yang et al., 2022) | DeBERTa-V2 | 224 | 32.2 | – | 16.8 | – | 24.7 | – |
| VideoLLaMA (Zhang et al., 2023a) | Vicuna-7B | 224 | 51.6 | 2.5 | 29.6 | 1.8 | 12.4 | 1.1 |
| LLaMA-Adapter (Gao et al., 2023) | LLaMA-7B | 224 | 54.9 | 3.1 | 43.8 | 2.7 | 34.2 | 2.7 |
| VideoChat (Li et al., 2023e) | Vicuna-7B | 224 | 56.3 | 2.8 | 45.0 | 2.5 | 26.5 | 2.2 |
| Video-ChatGPT (Maaz et al., 2023) | Vicuna-7B | 224 | 64.9 | 3.3 | 49.3 | 2.8 | 35.2 | 2.7 |
| LLaMA-VID (Li et al., 2023g) | Vicuna-7B | 224 | 69.7 | 3.7 | 57.7 | 3.2 | 47.4 | 3.3 |
| VideoChat2 (Li et al., 2023f) [CVPR'24] | Vicuna-7B | 224 | 70.0 | 3.9 | 54.1 | 3.3 | 49.1 | 3.3 |
| MiCo-Chat-7B | Vicuna-7B | 224 | **73.7** | **4.1** | **60.1** | **3.6** | **50.1** | **3.3** |

on ClothoV2, and 41.0% on AudioCaps, while in audio captioning, it achieves 49.6% on ClothoV1, and 50.8% on ClothoV2, all outperforming previous best results. For text-to-video retrieval, MiCo sets new SOTA performances with metrics of 64.3% R@1 on MSRVTT and 81.3% on VATEX, and in video-audio caption, it achieves impressive performances of 79.3% on MSRVTT, 197.8% on YouCook2, and 62.8% on VALOR-32K. Finally, in video-audio QA, MiCo also delivers superior performances of 50.4% on MSRVTT, 79.9% on MUSIC, and 51.0 on ANET. *These results collectively highlight MiCo's exceptional and versatile capabilities in cross-modal comprehension and reasoning tasks, establishing it as a promising direction in this field.*

## 4.4 EVALUATION ON MULTIMODAL UNDERSTANDING WITH LARGE LANGUAGE MODELS

**MiCo highlights its Omni-modal Zero-shot Comprehension and Reasoning Abilities**. Beyond traditional caption, retrieval, and QA tasks, we also evaluate the abilities of MiCo aligned with LLMs for zero-shot multimodal QA. We use ChatBridge (Zhao et al., 2023) as our baseline and Vicuna-7B as the large language model for each modality. As shown in Table 4, 5, and 6, MiCo-Chat-7B shows outstanding performances across both Vision LLMs and Multimodal LLMs. It directly delivers outstanding performances on the SQA (77.5%), MMB (70.4%), MMVet (49.3%), and SEED (69.2%) benchmarks while another 4 competitive performances. Besides, MiCo-Chat-7B also delivers

Table 7: **Ablation Study** on pretraining modalities, data scale, pretraining process, and parameters. Our default setting is to pretrain a base model for 30k steps with 10M data using all objective functions and evaluate it on the MSRVTT, VATEX, DIDEMO, MSVD, AudioCaps, ClothoV2, COCO, and Flicker datasets for retrieval tasks.

| Model | Factors | Video | | | | Audio | | Image | | Average |
|---|---|---|---|---|---|---|---|---|---|---|
| | | MSRVTT(VA) | VATEX(VA) | DIDEMO(VA) | MSVD(V) | AudioCaps(A) | ClothoV2(A) | COCO(I) | Flickr(I) | |
| **Pretraining Modalities** | | | | | | | | | | |
| (a) | I | 39.7 | 57.3 | 38.4 | 39.7 | 10.2 | 4.4 | 50.2 | 75.7 | 39.4 |
| (b) | I+3D | 42.0 | 58.5 | 38.1 | 40.1 | 10.8 | 4.2 | 51.2 | 76.9 | 40.2 |
| (c) | I+A | 37.6 | 56.2 | 30.8 | 36.2 | 22.0 | 14.5 | 46.8 | 71.0 | 39.4 |
| (d) | I+V | 41.7 | 60.9 | 39.2 | 42.6 | 12.2 | 5.1 | 51.3 | 77.0 | 41.3 |
| (e) | I+V+A | 42.2 | 61.1 | 40.1 | 41.2 | 23.4 | 15.4 | 48.7 | 74.2 | 43.2 |
| (f) | I+V+A+3D | **45.7** ↑6.0 | **64.0** ↑6.7 | **42.7** ↑4.3 | **42.8** ↑3.1 | **24.6** ↑14.4 | **15.9** ↑11.5 | 49.9 | **77.1** ↑1.4 | **45.3** ↑5.9 |
| **Data Scale** | | | | | | | | | | |
| (h) | 1M | 44.2 | 63.2 | 40.1 | 40.7 | 21.9 | 11.2 | 48.2 | 77.5 | 43.4 |
| (i) | 10M | 45.7 | 64.0 | 42.7 | 42.8 | 24.6 | 15.9 | 49.9 | 77.1 | 45.3 |
| (j) | 110M | 48.5 | 65.7 | 41.7 | 43.0 | 26.3 | 17.1 | 49.6 | 78.1 | 46.3 |
| (k) | 334M | **49.1** ↑4.9 | **66.3** ↑3.1 | **43.2** ↑3.1 | **44.1** ↑3.4 | **27.0** ↑5.1 | **17.5** ↑6.3 | **51.5** ↑3.3 | **80.9** ↑3.4 | **47.5** ↑4.1 |
| **Pretraining Process** | | | | | | | | | | |
| (l) | $\mathcal{L}_{Con}$ | 40.1 | 57.4 | 39.1 | 41.4 | 23.1 | 14.4 | 47.4 | 73.7 | 42.1 |
| (m) | $\mathcal{L}_{Con} + \mathcal{L}_{Match}$ | 43.9 | 61.4 | 38.0 | 41.6 | 23.6 | 15.5 | 48.8 | 74.3 | 43.4 |
| (n) | $\mathcal{L}_{Con} + \mathcal{L}_{Match} + \mathcal{L}_{Gen}$ | **45.7** ↑5.6 | **64.0** ↑6.6 | **42.7** ↑3.6 | **42.8** ↑1.4 | **24.6** ↑1.5 | **15.9** ↑1.5 | **49.9** ↑2.5 | **77.1** ↑3.4 | **45.3** ↑3.2 |
| **Model Scale** | | | | | | | | | | |
| (o) | Base-86M | 45.7 | 64.0 | 42.7 | 42.8 | 24.6 | 15.9 | 49.9 | 77.1 | 45.3 |
| (p) | Large-331M | 58.2 | 72.0 | 57.2 | 52.8 | 31.6 | 18.7 | 60.8 | 87.5 | 54.9 |
| (q) | Giant-1.3B | **62.5** ↑16.8 | **79.9** ↑15.9 | **61.1** ↑18.4 | **56.0** ↑13.2 | **37.4** ↑12.8 | **20.8** ↑4.9 | **67.1** ↑17.2 | **90.7** ↑13.6 | **59.4** ↑14.1 |

significantly impressive performances on both zero-shot caption and QA tasks on audio and video modalities, where **MiCo-Chat-7B achieves 6 new SOTA performances** including Clotho Caption, AQA, MSVD-QA, MSRVTT-QA, ActivityNet-QA. *These results are important proof that the MiCo pretraining paradigm shows a promising direction in developing large omni-modal models.*

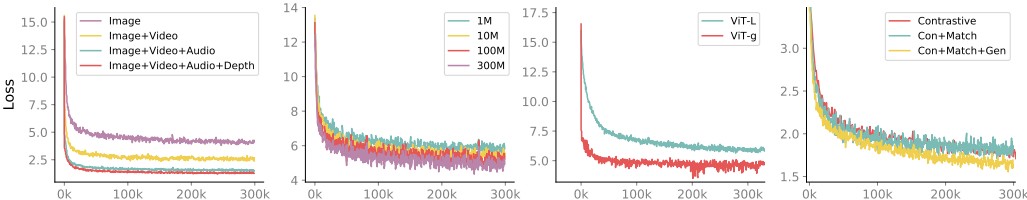

Figure 6: **Scalability of MiCo**. Loss curves under scaling factors (modality, data, parameters, process) settings.

## 4.5 ABLATION STUDY: SCALABILITY

**Scaling Modalities**. From (a) to (f), we gradually scale up input modalities. In Figure 6, all modalities (I+V+A+3D) achieves the highest scores, highlighting the importance and effectiveness of MiCo for diverse multimodal inputs.

**Scaling Multimodal Data.** From (h) to (k) in Table 7, we investigate the impact of the omni-modal data scale from 1M to 334M. It proves that the MiCo has great potential for further scaling.

**Pretraining Objectives.** From (l) to (n), we analyze the impact of each pretraining objective. The combination of contrastive, matching, and generative losses ($\mathcal{L}_{Con} + \mathcal{L}_{Match} + \mathcal{L}_{Gen}$) yields the best performance, demonstrating the value of multiple complementary objectives.

**Scaling Parameters.** From (o) to (q), we assess the effect of model size. Larger models, particularly the Giant-1.3B, show superior performance, confirming that increasing model parameters with MiCo enhances learning and generalization abilities across diverse modalities.

## 5 CONCLUSION AND LIMITATION

In this paper, we propose a novel framework, termed MiCo, to train foundation models with enhanced visual perception abilities and omni-modal capacities. With experiments on a reasonably large scale of both model and data, we conclude that the key to omni-modal learning is to simulate the multimedia cognition process of the human brain. In MiCo, we use image, depth, and normal maps to simulate the fundamental visual perception ability, distance spatial awareness, and geometry awareness of human visual cognition. In addition, captions, audio, and video provide prior knowledge, auditory perception, and spatial-temporal awareness. In future work, we plan to enhance our joint pretraining by incorporating additional modalities, including optical flow, IMU data, and event files, *etc*. We believe MiCo is an important attempt to simulate the multimedia cognition of human brains, and we expect it could inspire future works to develop more powerful omni-modal foundation models.

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
