# Appendix

## A  SUMMARY

The appendix is organized as follows:

- § B detailed training and evaluation settings of our models including hyper-parameters regarding models and optimizers.
- § C presents a comprehensive introduction on the datasets we use for evaluation and their corresponding metrics.

## B  TRAINING CONFIGURATION

### B.1  PRETRAINING SETTINGS

We detail the specific pretraining configurations of MiCo, focusing on the multi-dataset joint training corpora, the dataset mix ratios for each corpus, and the learning objectives for each corpus. To improve data quality, we employed a trained vision captioner to generate new captions for the CC4M datasets, replacing the original captions. Although MiCo has only been trained for 300,000 steps, it has already demonstrated outstanding performance on various downstream tasks. We anticipate that further increasing the number of training steps will significantly enhance the model's capabilities.

The pretraining of MiCo involves a combination of different datasets, each contributing uniquely to the model's learning process. With a parameter size of 1.0 billion and a sample size of 334 million, the model utilizes a diverse training corpus to achieve its results.

1. VAST-27M: This dataset contributes 324 million samples to the training process. With a batch size of 2048, the model undergoes 160,000 steps, completing one epoch.

2. VALOR-1M: In this dataset, 1 million samples are used with a batch size of 1024. The training spans 70,000 steps, which equates to approximately 71.7 epochs.

3. WavCaps, CC4M, and WebVid-2.5M: These datasets are combined, contributing 9 million samples in total. The batch size for this combined dataset is 1024, and the model is trained over 70,000 steps, resulting in 8.0 epochs.

The careful selection and combination of these datasets, along with the application of new, high-quality captions for the CC4M datasets, enhance the training efficiency and the quality of the learned representations.

### B.2  FINE-TUNING SETTINGS

We detail the downstream task finetuning settings, specifying the learning rate, batch size, epoch, training objectives, and resolution. The configurations also include the number of sampled video frames or audio clips used in training and testing phases. Here are the comprehensive settings:

**Retrieval Tasks (RET)**

- **Image-Text Modality**
  - **MSCOCO**: Learning rate of 1e-5, batch size of 256, 5 epochs, with the objective for retrieval, and a resolution of 384.
  - **Flickr**: Learning rate of 1e-5, batch size of 256, 5 epochs, with the objective for retrieval, and a resolution of 384.
- **Audio-Text Modality (A-T)**
  - **ClothoV1/V2**: Learning rate of 2e-5, batch size of 64, 10 epochs, with the objective for retrieval, using 3 audio clips during both training and testing.

- **AudioCaps**: Learning rate of 2e-5, batch size of 64, 10 epochs, with the objective for retrieval, using 1 audio clip during both training and testing.

- **Multi-modal (MM)**

  - **MSRVTT**: Learning rate of 2e-5, batch size of 64, 3.6 epochs, with the objective for retrieval, using 8 video frames during training and 16 during testing, with a resolution of 224.
  - **YouCook2**: Learning rate of 3e-5, batch size of 64, 30 epochs, with the objective for retrieval, using 8 video frames during training and 16 during testing, with a resolution of 224.
  - **VALOR-32K**: Learning rate of 2e-5, batch size of 64, 10 epochs, with the objective for retrieval, using 8 video frames during both training and testing, with a resolution of 224.
  - **VATEX**: Learning rate of 2e-5, batch size of 64, 2.5 epochs, with the objective for retrieval, using 8 video frames during training and 16 during testing, with a resolution of 224.
  - **DiDeMo**: Learning rate of 2e-5, batch size of 64, 40 epochs, with the objective for retrieval, using 8 video frames during training and 32 during testing, and 2 audio clips during both training and testing, with a resolution of 224.
  - **ANET**: Learning rate of 2e-5, batch size of 64, 20 epochs, with the objective for retrieval, using 8 video frames during training and 32 during testing, and 2 audio clips during both training and testing, with a resolution of 224.

**Captioning Tasks (CAP)**

- **Image-Text Modality**

  - **MSCOCO**: Learning rate of 1e-5, batch size of 64, 5 epochs, with the objective for caption, and a resolution of 480.
  - **MSCOCO(SCST)**: Learning rate of 2.5e-6, batch size of 64, 2.5 epochs, with the objective for caption, and a resolution of 480.

- **Audio-Text Modality (A-T)**

  - **ClothoV1/V2**: Learning rate of 2e-5, batch size of 64, 10 epochs, with the objective for caption, using 3 audio clips during both training and testing.
  - **AudioCaps**: Learning rate of 2e-5, batch size of 64, 10 epochs, with the objective for caption, using 1 audio clip during both training and testing.

- **Multi-modal (MM)**

  - **MSRVTT**: Learning rate of 2e-5, batch size of 128, 10 epochs, with the objective for caption, using 8 video frames during both training and testing, with a resolution of 224.
  - **YouCook2**: Learning rate of 3e-5, batch size of 64, 30 epochs, with the objective for caption, using 8 video frames during training and 16 during testing, with a resolution of 224.
  - **VALOR-32K**: Learning rate of 1e-5, batch size of 64, 10 epochs, with the objective for caption, using 8 video frames during training and 12 during testing, with a resolution of 224.

**Question Answering Tasks (QA)**

- **Visual-Text Modality (Vis)**

  - **MSVD-QA**: Learning rate of 1e-5, batch size of 64, 10 epochs, with the objective for QA, using 8 video frames during training and 14 during testing, with a resolution of 224.
  - **TGIF-FrameQA**: Learning rate of 2e-5, batch size of 64, 10 epochs, with the objective for QA, using 4 video frames during both training and testing, with a resolution of 224.
  - **VQAv2**: Learning rate of 2e-5, batch size of 128, 20 epochs, with the objective for QA, and a resolution of 384.

**Table 8: Detailed training configurations of MiCo for multimodal learning.** Apart from the configurations shown in the table, for image tasks, we use random left-right flipping, random resized crop, color jitter of 0.4, Auto-augment, and no repeated augmentation for every model.

| settings | Image | | Audio | | Video | | Depth & Normal Map | |
|---|---|---|---|---|---|---|---|---|
| | ViT-L | ViT-g | ViT-L | ViT-g | ViT-L | ViT-g | ViT-L | ViT-g |
| Input Shape | 224 | 224 | 224 | 224 | 224 | 224 | 224 | 224 |
| batch size | 4096 | 512 | 4096 | 512 | 4096 | 512 | 4096 | 512 |
| optimizer | AdamW | AdamW | AdamW | AdamW | AdamW | AdamW | AdamW | AdamW |
| LR | $4\times10^{-3}$ | $5\times10^{-5}$ | $4\times10^{-3}$ | $5\times10^{-5}$ | $4\times10^{-3}$ | $5\times10^{-5}$ | $4\times10^{-3}$ | $5\times10^{-5}$ |
| LR schedule | cosine | cosine | cosine | cosine | cosine | cosine | cosine | cosine |
| weight decay | 0.05 | $1\times10^{-8}$ | 0.05 | $1\times10^{-8}$ | 0.05 | $1\times10^{-8}$ | 0.05 | $1\times10^{-8}$ |
| warmup epochs | 5 | 0 | 5 | 0 | 5 | 0 | 5 | 0 |
| epochs | 90 | 30 | 90 | 30 | 90 | 20 | 90 | 20 |
| mixup alpha | 0.8 | 0.0 | 0.8 | 0.0 | 0.8 | 0.0 | 0.8 | 0.0 |
| cutmix alpha | 1.0 | 0.0 | 1.0 | 0.0 | 1.0 | 0.0 | 1.0 | 0.0 |
| erasing prob. | 0.25 | 0.25 | 0.25 | 0.25 | 0.25 | 0.25 | 0.25 | 0.25 |
| dropout rate | 0.1 | 0.2 | 0.1 | 0.2 | 0.1 | 0.3 | 0.2 | 0.3 |

**Algorithm 1** Multimodal Context Pretraining Algorithm, PyTorch-like

```python
def train(video_pixels=None, image_pixels=None, depth_pixels=None, audio_spectrograms=
    None):
  # Get Mixed Data
  modal_inputs = [video_pixels, image_pixels, depth_pixels, audio_spectrograms]
  modal_captions = [video_captions, image_captions, depth_captions, audio_captions]

  # Extract Features
  modal_feats = [self.encoder(modal) for modal in modal_inputs if modal is not None]
  multimodal_feats = torch.cat(modal_feats)
  concatenated_captions = ''.join(modal_captions)
  text_feats = self.text_encoder(concatenated_captions)

  # Losses
  contra_loss = Contrasive_Loss(multimodal_feats, text_feats)
  matching_loss = Matching_Loss(modal_captions, multimodal_feats)
  gen_loss = Generation_Loss(modal_captions.mask(0.6), multimodal_feats)

  # Total Loss
  loss = contra_loss + matching_loss + gen_loss

return loss
```

- **Multi-modal (MM)**

    - **MSRVTT-QA**: Learning rate of 2e-5, batch size of 64, 4.5 epochs, with the objective for QA, using 8 video frames and 1 audio clip during both training and testing, with a resolution of 224.

    - **MUSIC-AVQA**: Learning rate of 2e-5, batch size of 64, 20 epochs, with the objective for QA, using 8 video frames and 2 audio clips during both training and testing, with a resolution of 224.

    - **ANET-QA**: Learning rate of 2e-5, batch size of 64, 10 epochs, with the objective for QA, using 8 video frames during training and 16 during testing, and 2 audio clips during both training and testing, with a resolution of 224.

These settings have been optimized to balance efficiency and performance, even though most hyperparameters are not precisely tuned.

For evaluation purposes, we employ different strategies tailored to specific tasks:

1. Retrieval Tasks: All candidates are initially ranked using Omni-modal Contrastive Loss. Following this, the Top-50 candidates undergo a reranking process through the Omni-modal Matching Process.

**Algorithm 2** Dataset Split Algorithm

```python
import pandas as pd
from sklearn.model_selection import train_test_split

# Assume 'data' is a DataFrame containing the full dataset with columns ['category', '
    vision_caption', 'audio_caption', 'depth', 'normal', 'subtitle']
# Adding an 'index' column to keep track of the original indices
data['index'] = data.index

# Define the sizes of each subset
subset_sizes = [1e6, 1e7, 1.1e7, 3.34e7]

# Function to create stratified samples
def create_subset(data, size):
    subset, _ = train_test_split(data, train_size=size, stratify=data['category'],
        random_state=42)
    return subset

# Creating subsets
subset_1M = create_subset(data, 1e6)
subset_10M = create_subset(data, 1e7)
subset_110M = create_subset(data, 1.1e7)
subset_334M = create_subset(data, 3.34e7)

# Reset index for each subset
subset_1M.reset_index(drop=True, inplace=True)
subset_10M.reset_index(drop=True, inplace=True)
subset_110M.reset_index(drop=True, inplace=True)
subset_334M.reset_index(drop=True, inplace=True)
```

2. Captioning Tasks: Beam search with a beam size of 3 is utilized to generate captions, ensuring a comprehensive exploration of possible outputs.

3. Question Answering (QA) Tasks: These are treated as open-ended generative problems. Questions are used as prefixes, and answers are generated without any constraints, allowing for flexible and contextually appropriate responses.

For comparisons with state-of-the-art (SOTA) models and ablation studies, we use the following evaluation metrics: 1) Retrieval Tasks: Recall@1. 2) Captioning Tasks: CIDEr. 3) QA Tasks: Accuracy (Acc) These metrics provide a comprehensive assessment of the model's performance across different types of tasks.

## C DATASETS AND METRICS

**Dataset Split** To split the mix of datasets into subsets of 1M, 10M, 110M, and 334M video clips while preserving its diversity and quality, we employed a proportional stratified sampling method. Initially, the dataset, which spans over 15 categories (including music, gaming, education, entertainment, and animals) and includes vision, audio, depth, normal maps, and text modalities, was organized and labeled. Stratified random sampling was then used to ensure each subset accurately reflected the distribution of categories and modalities present in the full dataset. This method involved selecting samples proportionally from each category to maintain representative distributions. The vision and audio captions were also kept proportional in length and quantity, ensuring that each subset retained the comprehensive characteristics of the original dataset.

### C.1 SINGLE-MODALITY EVALUATION DETAILS

**Text**. The MMLU (Massive Multitask Language Understanding) benchmark is designed to evaluate the multitask accuracy of language models across 57 diverse tasks, including subjects like mathemat-

ics, history, and biology. It assesses models' abilities to generalize and apply knowledge in various domains, providing a comprehensive measure of text understanding and reasoning skills.

**Image**. We conduct experiments on ImageNet-1K (Deng et al., 2009), a dataset comprising approximately 1.3 million images across 1,000 categories. In line with common practices (Wang et al., 2021a; Liu et al., 2021; 2022; Ding et al., 2023), base-scale models are trained for 300 epochs. Large-scale models undergo pre-training on ImageNet-22K, which includes 14.2 million images, for 90 epochs, followed by fine-tuning on ImageNet-1K for an additional 20 epochs.

**Thermal and Hyperspectral data understanding**. We conduct experiments on infrared image recognition using the RegDB dataset, X-ray scan analysis with the Chest X-Ray dataset (Rahman et al., 2020), and hyperspectral data recognition using the Indian Pine dataset[2].

**Depth**. The NYU Depth Dataset (NYU-D) comprises RGB and depth image pairs captured from indoor scenes. It includes 1,449 densely labeled pairs for training and testing, along with over 400,000 unlabeled frames.

**Audio**. For audio recognition, Audioset-2M dataset comprises over 2 million human-labeled 10-second audio clips drawn from YouTube videos. It covers a wide range of 527 sound event classes, providing a comprehensive resource for training and evaluating audio event detection and classification models.

**Video**. The Kinetics-700 dataset contains 700,000 video clips covering 700 human action classes, used for action recognition tasks. The MSR-VTT dataset includes 10,000 video clips paired with multiple textual descriptions, supporting video captioning, retrieval, and content understanding research.

**Time-series**. Global Weather Forecasting (Wu et al., 2023) includes global, regional, and Olympics data from NCEI and CMA, comprising hourly weather measurements from thousands of stations. Evaluation involved splitting data into training, validation, and test sets (7:1:2) using MSE and MAE metrics.

**Graph**. PCQM4M-LSC dataset is a large-scale collection of 4.4 million organic molecules, each with up to 23 heavy atoms and associated quantum-mechanical properties. Aimed at predicting molecular properties through machine learning, this dataset is highly relevant for applications in drug discovery and material science.

**Tabular**. The fraud dataset comprises transaction records, including features like transaction amount, location, time, and user information. It is designed for machine learning models to detect fraudulent activities. This dataset is crucial for developing and testing algorithms to enhance security in financial systems and reduce economic losses due to fraud.

**IMU**. The Ego4D dataset includes inertial measurement unit (IMU) data captured from wearable devices, providing detailed motion and orientation information. This dataset supports research in human activity recognition, augmented reality, and robotics, offering comprehensive insights into human movements and interactions with the environment.

## C.2 CROSS-MODALITY EVALUATION DETAILS

We evaluated MiCo across several well-known downstream datasets, including MSRVTT, VATEX, YouCook2, VALOR-32K, MSVD, DiDeMo, ActivityNet Caption, TGIF, MUSIC-AVQA, Clotho, AudioCaps, MSCOCO, Flickr30K, and VQAv2. The specific train/validation/test splits for these benchmarks are detailed below:

---

[2]https://github.com/danfenghong/IEEE_TGRS_SpectralFormer/blob/main/data/IndianPine.mat

## RETRIEVAL TASKS

### AUDIO-TEXT MODALITY (A-T)

- **ClothoV1** (Drossos et al., 2020): This dataset includes 2,893 audio clips for training and 1,045 for validation. The corresponding captions number 14,465 for training and 5,225 for validation.

- **ClothoV2** (Drossos et al., 2020): Contains 3,839 audio clips for training and 1,045 for validation, with 19,195 captions for training and 5,225 for validation.

- **AudioCaps** (Kim et al., 2019): Comprises 49,291 audio clips for training, 428 for validation, and 816 for testing, along with 49,291 captions for training, 2,140 for validation, and 4,080 for testing.

### VIDEO-TEXT MODALITY (V-T)

- **MSRVTT** (Xu et al., 2016a): Comprises 10K video clips and 200K captions, spanning diverse topics such as human activities, sports, and natural landscapes. We evaluate text-to-video retrieval, video captioning, and video QA using this dataset. Contains 9,000 videos for training and 1,000 for testing, with 180,000 captions for training and 1,000 for testing.

- **YouCook2** (Zhou et al., 2018): Comprises 14K video clips extracted from 2K instructional cooking videos on YouTube. Each video features multiple actions performed by chefs, along with corresponding textual descriptions and temporal annotations. Includes 10,337 videos for training and 3,492 for validation, with matching captions.

- **VALOR-32K** (Chen et al., 2023a): An audiovisual video-language benchmark containing 32K 10-second video clips sourced from AudioSet (Gemmeke et al., 2017). Each clip includes annotations with captions that describe both the visual and audio content. Consists of 25,000 videos for training, 3,500 for validation, and 3,500 for testing, each with corresponding captions.

- **DiDeMo** (Anne Hendricks et al., 2017): Comprises 10K long-form videos sourced from Flickr, with each video annotated with four short sentences in temporal order. For this benchmark, we concatenate these short sentences and evaluate 'paragraph-to-video' retrieval, using the official split. Features 8,394 videos for training, 1,065 for validation, and 1,003 for testing, along with their captions.

- **ActivityNet (ANET)** (Krishna et al., 2017): Includes 20K long-form videos (average length of 180 seconds) from YouTube, accompanied by 100K captions. We evaluate text-to-video retrieval and video QA on this dataset. Comprises 10,009 videos for training and 4,917 for testing, with corresponding captions.

- **LSMDC** (Rohrbach et al., 2017): Contains 101,046 videos for training, 7,408 for validation, and 1,000 for testing, with corresponding captions.

## CAPTIONING TASKS

### AUDIO-TEXT MODALITY (A-T)

- **ClothoV1** (Drossos et al., 2020): This dataset includes 2,893 audio clips for training and 1,045 for validation. The corresponding captions number 14,465 for training and 5,225 for validation.

- **ClothoV2** (Drossos et al., 2020): Contains 3,839 audio clips for training and 1,045 for validation, with 19,195 captions for training and 5,225 for validation.

- **AudioCaps** (Kim et al., 2019): Comprises 49,838 audio clips for training, 495 for validation, and 975 for testing, along with 49,438 captions for training, 2,475 for validation, and 4,875 for testing.

VIDEO-TEXT MODALITY (V-T)

- **MSRVTT** (Xu et al., 2016a): Contains 6,513 videos for training, 497 for validation, and 2,990 for testing, with 130,260 captions for training, 9,940 for validation, and 59,800 for testing.
- **YouCook2** (Zhou et al., 2018): Includes 10,337 videos for training and 3,492 for validation, with matching captions.
- **VALOR-32K** (Chen et al., 2023a): Consists of 25,000 videos for training, 3,500 for validation, and 3,500 for testing, each with corresponding captions.
- **VATEX** (Wang et al., 2019): Consists of 41,250 video clips sourced from the Kinetics-600 dataset (Kay et al., 2017), accompanied by 825,000 sentence-level descriptions. Contains 25,991 videos for training, 3,000 for validation, and 6,000 for testing, with 259,910 captions for training, 30,000 for validation, and 60,000 for testing.

QUESTION ANSWERING (QA) TASKS

VIDEO-TEXT MODALITY (V-T)

- **MSRVTT-QA** (Xu et al., 2017): Contains 6,513 videos for training, 497 for validation, and 2,990 for testing, with 158,581 QA pairs for training, 12,278 for validation, and 72,821 for testing.
- **MUSIC-AVQA** (Li et al., 2022): An audiovisual video QA benchmark containing over 45K Q-A pairs, covering 33 different question templates across various modalities and question types. Includes 9,277 videos for training, 3,815 for validation, and 6,399 for testing, with 32,087 QA pairs for training, 4,595 for validation, and 9,185 for testing.
- **ANET-QA** (Yu et al., 2019a): Comprises 3,200 videos for training, 1,800 for validation, and 800 for testing, with 32,000 QA pairs for training, 18,000 for validation, and 8,000 for testing.

IMAGE-BASED TASKS

- **MSCOCO** (Lin et al., 2014): Comprises 123K images, each paired with 5 annotated captions. We evaluate text-to-image retrieval and image captioning on this dataset.
- **Flickr30K** (Plummer et al., 2015): Contains 31K images, each paired with five descriptive captions. This dataset is widely used for evaluating image captioning and text-to-image retrieval tasks.

VISUAL QUESTION ANSWERING

- **VQAv2** (Goyal et al., 2017a): A large-scale Visual Question Answering dataset comprising over 265K images and 1.1M questions, designed to improve the balance of answer types per question. This dataset is used to evaluate models' abilities to understand and reason about visual content by providing accurate answers to questions based on the images.