# OpenReview forum: "Scaling Omni-modal Pretraining with Multimodal Context: Advancing Universal Representation Learning Across Modalities"
_ICLR.cc/2025/Conference — ICLR 2025 Conference Withdrawn Submission_

### Official Review · Reviewer_zikJ · 2024-10-27

**Soundness:** 1
**Presentation:** 2
**Contribution:** 3
**Rating:** 3
**Confidence:** 4

**Summary:**

The paper introduces MiCo, a pretraining framework designed to enable what the authors call “omni-modal intelligence” by integrating diverse data types (such as text, image, audio, video, and other image-representable modalities such as depth and normal maps and audio spectrograms) into a universal embedding space by CLIP-like pretraining. MiCo demonstrates substantial improvements over other baselines models that work with fewer modalities. The model architecture leverages the idea of CLIP alignment and applies it to even more image types than just natural images.

**Strengths:**

**Extensive Benchmarking and good results on those benchmarks(*):** The paper tests MiCo’s performance across an immense number of benchmarks, which speaks to MiCo’s robustness and adaptability across diverse tasks compared to baselines that handle fewer modalities (*).

(*): see W3.

**Weaknesses:**

**W1. Overstated terminology:** Why introduce the term "omni-modal" (omni = all, implying all modalities) when "multi-modal" (multi = many, thus many modalities) is well-established in the literature? "Omni" seems an overstatement here, as the system doesn’t cover all possible modalities—and realistically, it never will. The concept of “all modalities” is a bit of a moving target; as technology advances, new sensors and sources of data continuously emerge. The term "omni-modal" risks muddling the field by adding jargon without adding clarity. Multimodality already captures the idea of incorporating as many modalities as possible, and researchers often use "massively multimodal" [1, 2] to describe systems integrating a particularly broad set of inputs. Introducing "omni-modal" doesn’t seem necessary and only complicates terminology.

**W2. Limited technical novelty:** Since all the additional modalities in this (overstating) “omni” architecture are ultimately represented as images, it’s not surprising that a Vision Transformer (ViT), much like in the original CLIP, can process them effectively. This approach feels more like a straightforward extension of CLIP to image-representable data than a true “omni-modal” architecture. A genuinely omni-modal system would likely handle a broader variety of data types (like olfactory signals, chemical sensors, tactile feedback, or proprioceptive data) rather than sticking to modalities that can be easily represented as images. Thus, it is hard to see how the innovations of this paper are living up to the standards this paper sets.

**W3. Limited comparisons to comparable work:** The paper primarily benchmarks against two-modal baselines, missing the opportunity to position itself meaningfully within the space of massively multimodal architectures (to which the paper would refer as "omni-modal"). For fair comparison, the paper should include baselines like [1] and [2] at the very least. Additionally, in the “More-Modality Pretraining” section (L151) of Related Work, there’s no mention of these comparable studies, leaving this section incomplete and missing key references.

And, curiously, why is it called “more-modality pretraining” (honest naming) here in the related work instead of “omni-modality” (over-statement) as it is elsewhere? Consistent terminology would clarify the scope and impact of the work.

[1] https://arxiv.org/pdf/2312.06647

[2] https://arxiv.org/abs/2406.09406

**Questions:**

**Questions:**
* How does MiCo handle data with inherent noise or high variance across modalities (e.g., audio in a noisy environment)?
* The method section, paper and supplement lack any description about how the graph, tabular and IMU modalities are fed to the ViT to fit into the architecture. How is this done?

**Typos:**
* L092: Great introduction writing would explain in one sentence what this “novel generative reasoning method” does.
* L092: “In Transformer” should be “In the Transformer architecture” / “In Transformers”
* L316: 4. “Experiment” should be “Experiments” as this paper has not just one experiment.
* L475: “Conclusion and Limitation” should be “Conclusions and Limitations” as any paper has more than one limitation. The plural to conclusion would be voluntary here.

---

### Official Review · Reviewer_kLFx · 2024-11-03

**Soundness:** 3
**Presentation:** 3
**Contribution:** 2
**Rating:** 5
**Confidence:** 4

**Summary:**

The paper proposes a multi-modal framework that integrates various modalities, including image, video, audio, and depth information, to enhance performance on several downstream tasks. the approach achieves state-of-the-art results across multiple benchmarks.

**Strengths:**

Diverse Modal Representation: The paper successfully explores multiple modalities, which is essential for advancing omni-modal understanding.
Performance on Benchmarks: The method achieves state-of-the-art results on a majority of the tested benchmarks, indicating strong performance in specific tasks.
The paper includes detailed ablation experiments analyzing the model structure, modality selection, as well as data and parameter scaling.

**Weaknesses:**

1. Insufficient Rationale for Modality Efficacy: The introduction of new modalities lacks a strong theoretical justification. The performance boost from the inclusion of 3D information is particularly puzzling without a clear explanation.
2. Limited Evaluation of Omni-modality: The experiments focus on single and cross-modal capabilities but do not sufficiently address omni-modal understanding, there are some omni-modality benchmarks available, such as Omni×R(https://arxiv.org/pdf/2410.12219), Omnibench(https://arxiv.org/abs/2409.15272).
3. Generalizability Concerns: While the framework performs well on specific tasks, there is insufficient evidence to support its generalizability across diverse applications and domains.
4. Lack of Code and Data Availability: The paper mentions open-source code and data, but doesn't see links to open source

**Questions:**

1. Why does the introduction of 3D information improve performance? The paper only demonstrates the superiority of introducing 3D modal information, but does not provide an in-depth discussion of the reasons.
2. The paper uses a ViT architecture to represent and encode multiple types of information, including images, videos, audio, and 3D. How are the representations of different modalities aligned in a single ViT space without conflict?
3. Some formatting errors: There are formatting issues present, such as the incorrect reference to the backbone in Table 4 (should be Vicuna-7B).

---

### Official Review · Reviewer_78aK · 2024-11-04

**Soundness:** 4
**Presentation:** 3
**Contribution:** 3
**Rating:** 5
**Confidence:** 4

**Summary:**

This paper presents a new pre-training framework for training high-quality multi-modal universal representations (omni-modal intelligence). The so-called MiCo (Multimodal Context) system scales well with respect to number of modalities as well as data volume. The core idea behind the approach is simple, which leverages off-the-shelf pre-trained models to "convert" information from one modality to another modality. For example, from a regular video dataset (HD-VILA), the authors enrich it by adding video frame captions, audio, depth maps and normal maps. With the synthetic paired multi-modal datasets, the authors designed some mixed objectives based on contrastive learning and masked-generation loss over cross-modality inputs. The authors benchmarked the pre-trained model on various single-modality and cross-modality tasks, showing competitive performance over existing state-of-the-art methods.

**Strengths:**

* Originality

This paper presents a simple yet scalable approach to craft synthetic paired data in multi-modality setup. The pre-training method is a natural extension and combination from existing work in text-only and vision-language MLM (masked modeling and contrastive learning).

* Quality

The authors have done extensive experiments over multiple existing tasks, showing the performance gain of the proposed approach.

* Clarity

The paper is relatively easy to follow and the key idea is well-demonstrated with equations and figures.

* Significance

The work is targeting an important research problem (omnimodal intelligence) and the proposed approach is simple enough to be adapted by other work.

**Weaknesses:**

* The key techinique behind the dataset collection is to enrich existing video datasets with aligned, cross-modality representations, which is achieved by leveraging off-the-shelf pre-trained models, e.g. captions to video frames and audio. The quality of the multi-modal datasets is directly affected by the selected models. However, this part is only briefly mentioned in Section 3.1. More ablations or discussions would be beneficial regarding this part, as I presume this affects the generic applicablity of the proposed dataset collection approach, i.e. whether the potential distribution shift between the pre-trained model checkpoint and the source video dataset would affect the quality of the generated dataset.

* The proposed model demonstrate impressive performance on many benchmarks (setting new SoTA scores) but more careful analysis probably is needed, especially for some pretty "old" benchmarks that the data might have been indirectly seen by the model via the "data curation" process. More details about the evaluation procedures would be helpful.

* There exists quite some grammatical errors and typos (e.g. L181-182, L212, L216, etc.) The paper would benefit from more careful revisions.

**Questions:**

* For clarification, after constructing the paired dataset, is the model components (encoders etc.) trained from scratch or initialized from some existing checkpoints? How many epochs after 30K or 300K steps on the 1M/10M dataset? Also, if a new modality is required to be added, is it necessary to re-train the entire system or incremental adjustment can be made?

* Also see my comments of dataset curation in weaknesses section.

---

### Note · Authors · 2024-11-13

I have read and agree with the venue's withdrawal policy on behalf of myself and my co-authors.